



# Molecular understanding of the suppression of new-particle formation by isoprene

Martin Heinritzi[1], Lubna Dada[2], Mario Simon[1], Dominik Stolzenburg[3], Andrea C. Wagner[1,4], Lukas Fischer[5], Lauri R. Ahonen[2], Stavros Amanatidis[6], Rima Baalbaki[2], Andrea Baccarini[7], Paulus S. Bauer[3], Bernhard Baumgartner[3], Federico Bianchi[2,8], Sophia Brilke[3], Dexian Chen[9], Randall Chiu[4], Antonio Dias[10,11], Josef Dommen[7], Jonathan Duplissy[2], Henning Finkenzeller[4], Carla Frege[7], Claudia Fuchs[7], Olga Garmash[2], Hamish Gordon[11,12], Manuel Granzin[1], Imad El Haddad[7], Xucheng He[2], Johanna Helm[1], Victoria Hofbauer[9], Christopher R. Hoyle[13], Juha Kangasluoma[2,8], Timo Keber[1], Changhyuk Kim[6,14], Andreas Kürten[1], Houssni Lamkaddam[7], Janne Lampilahti[2], Tiia M. Laurila[2], Chuan Ping Lee[7], Katrianne Lehtipalo[2], Markus Leiminger[5], Huajun Mai[6], Vladimir Makhmutov[15], Hanna Elina Manninen[11], Ruby Marten[7], Serge Mathot[11], Roy Lee Mauldin[2,16,17], Bernhard Mentler[5], Ugo Molteni[7], Tatjana Müller[1], Wei Nie[18], Tuomo Nieminen[19], Antti Onnela[11], Eva Partoll[5], Monica Passananti[2], Tuukka Petäjä[2], Joschka Pfeifer[1,11], Veronika Pospisilova[7], Lauriane L. J. Quéléver[2], Matti P. Rissanen[2], Clémence Rose[2,20], Siegfried Schobesberger[19], Wiebke Scholz[5], Kay Scholze[3], Mikko Sipilä[2], Gerhard Steiner[5], Yuri Stozhkov[15], Christian Tauber[3], Yee Jun Tham[2], Miguel Vazquez-Pufleau[3], Annele Virtanen[19], Alexander L. Vogel[1,11], Rainer Volkamer[4], Robert Wagner[2], Mingyi Wang[9], Lena Weitz[1], Daniela Wimmer[2], Mao Xiao[7], Chao Yan[2], Penglin Ye[9,21], Qiaozhi Zha[2], Xueqin Zhou[1,7], Antonio Amorim[10], Urs Baltensperger[7], Armin Hansel[5], Markku Kulmala[2,8,22], António Tomé[23], Paul M. Winkler[3], Douglas R. Worsnop[2,21], Neil M. Donahue[9], Jasper Kirkby[1,11] and Joachim Curtius[1]

[1]Institute for Atmospheric and Environmental Sciences, Goethe University Frankfurt, 60438 Frankfurt am Main, Germany

[2]Institute for Atmospheric and Earth System Research (INAR) / Physics, Faculty of Science, University of Helsinki, 00014 Helsinki, Finland

[3]Faculty of Physics, University of Vienna, 1090 Vienna, Austria

[4]Department of Chemistry & CIRES, University of Colorado at Boulder, Boulder, CO, 80309-0215, USA

[5]Institute for Ion and Applied Physics, University of Innsbruck, 6020 Innsbruck, Austria

[6]California Institute of Technology, Pasadena, CA 91125, USA

[7]Laboratory of Atmospheric Chemistry, Paul Scherrer Institute, 5232 Villigen, Switzerland

[8]Aerosol and Haze Laboratory, Beijing Advanced Innovation Center for Soft Matter Science and Engineering, Beijing University of Chemical Technology, Beijing, 100029, P.R. China

[9]Center for Atmospheric Particle Studies, Carnegie Mellon University, Pittsburgh, PA, 15213, USA

[10]CENTRA and FCUL, University of Lisbon, 1749-016 Lisbon, Portugal

[11]CERN, 1211 Geneva, Switzerland

[12]University of Leeds, Leeds LS2 9JT, United Kingdom

[13]Institute for Atmospheric and Climate Science, ETH Zurich, Switzerland

[14]Department of Environmental Engineering, Pusan National University, Busan 46241, Republic of Korea





35    [15]Lebedev Physical Institute, Russian Academy of Sciences, 119991, Moscow, Russia

[16]Department of Atmospheric and Oceanic Sciences, University of Colorado at Boulder, Boulder, CO 80309, USA

[17]Department of Chemistry, Carnegie Mellon University, Pittsburgh, PA 15213, USA

[18]Joint International Research Laboratory of Atmospheric and Earth System Sciences, School of Atmospheric Sciences, Nanjing University, Nanjing, 210023, China

40    [19]Department of Applied Physics, University of Eastern Finland, PO Box 1627, FI-70211 Kuopio, Finland

[20]Laboratory for physical meteorology, UMR6016, University Clermont Auvergne-CNRS, 63178, Aubière, France

[21]Aerodyne Research, Inc., Billerica, MA 01821, USA

[22]Helsinki Institute of Physics, University of Helsinki, 00014 Helsinki, Finland

[23]IDL-University of Beira Interior, Covilhã, Portugal

45

*Correspondence to:* Joachim Curtius, curtius@iau.uni-frankfurt.de



**Abstract**

Nucleation of atmospheric vapors produces more than half of global cloud condensation nuclei and so has an important influence on climate. Recent studies show that monoterpene ($C_{10}H_{16}$) oxidation yields highly-oxygenated products that can nucleate with or without sulfuric acid. Monoterpenes are emitted mainly by trees, frequently together with isoprene ($C_5H_8$), which has the highest global emission of all organic vapors. Previous studies have shown that isoprene suppresses new-particle formation from monoterpenes, but the cause of this suppression is under debate. Here, in experiments performed under atmospheric conditions in the CERN CLOUD chamber, we show that isoprene reduces the yield of highly-oxygenated dimers with 19 or 20 carbon atoms - which drive particle nucleation and early growth - while increasing the production of dimers with 14 or 15 carbon atoms. The dimers (termed $C_{20}$ and $C_{15}$, respectively) are produced by termination reactions between pairs of peroxy radicals ($RO_2\cdot$) arising from monoterpenes or isoprene. Compared with pure monoterpene conditions, isoprene reduces nucleation rates at 1.7 nm (depending on the isoprene/monoterpene ratio) and approximately halves particle growth rates between 1.3 and 3.2 nm. However, above 3.2 nm, $C_{15}$ dimers contribute to secondary organic aerosol and the growth rates are unaffected by isoprene. We further show that increased hydroxyl radical ($OH\cdot$) reduces particle formation in our chemical system rather than enhances it as previously proposed, since it increases isoprene derived $RO_2\cdot$ radicals that reduce $C_{20}$ formation. $RO_2\cdot$ termination emerges as the critical step that determines the HOM distribution and the corresponding nucleation capability. Species that reduce the $C_{20}$ yield, such as NO, $HO_2$ and as we show isoprene, can thus effectively reduce biogenic nucleation and early growth. Therefore the formation rate of organic aerosol in a particular region of the atmosphere under study will vary according to the precise ambient conditions.

## 1. Introduction

Nucleation of aerosol particles is observed in many environments, ranging from boreal forests to urban and coastal areas, from polar to tropical regions and from the boundary layer to the free troposphere (Kerminen et al., 2018). Gaseous sulfuric acid, ammonia (Kirkby et al., 2011), amines (Almeida et al., 2013) and, in coastal regions, iodine (Sipilä et al., 2016), were shown to contribute to nucleation. Additionally, a small fraction of the large pool of organic molecules in the atmosphere, namely highly-oxygenated organic molecules (HOMs), some of which possess extremely low vapor pressures, nucleate together with other precursors as well as on their own (Riccobono et al., 2014; Kirkby et al., 2016; Tröstl et al., 2016). This means nature is nucleating particles on a large scale without pollution, and this may have been especially pervasive in the pre-industrial atmosphere (Gordon et al., 2016). HOMs can be formed with molar yields in the single-digit percent range from the oxidation of monoterpenes ($C_{10}H_{16}$) with endocyclic C=C double-bonds (Kirkby et al., 2016; Ehn et al., 2014). Monoterpenes are emitted by a variety of trees in regions ranging from the tropics to northern latitudes, often reaching mixing ratios of tens to hundreds of parts per trillion by volume (pptv) (Jardine et al., 2015; Guenther et al., 2012). Isoprene is a hemiterpene ($C_5H_8$) emitted by broad-leaf trees and has the highest emissions of any biogenic organic compound, with concentrations reaching several parts per billion by volume (ppbv) in the Amazon rainforest and the southeastern United States despite high reactivity (Guenther et al., 2012; Martin et al., 2010; Lee et al., 2016). Numerous studies report suppression of nucleation in isoprene-rich environments, even if sufficient monoterpenes are present (Lee et al., 2016; Kanawade et al., 2011; Yu et al., 2014; Kiendler-Scharr et al., 2009; Kiendler-Scharr et al., 2012; Varanda Rizzo et al., 2018; Wimmer et al., 2018). This isoprene suppression effect has been demonstrated in carefully controlled chamber studies (Kiendler-Scharr et al., 2009; Kiendler-Scharr et al., 2012) and observed in isoprene-rich ambient locations (Kanawade et al., 2011; Lee et al., 2016; Yu et al., 2014). A recent study reported also a suppression of secondary organic aerosol (SOA) formation due to isoprene in an $OH\cdot$ dominated chamber experiment (McFiggans et al., 2019). In addition to observing suppression of particle formation by isoprene, earlier studies have proposed mechanisms to explain it. One possibility is $OH\cdot$ depletion by isoprene, which would reduce the oxidation rate of monoterpenes and thus supersaturation driving nucleation (Kiendler-Scharr et al., 2009; Kiendler-Scharr et al., 2012; McFiggans et al., 2019). However, $OH\cdot$ is observed to remain high and undisturbed in isoprene-rich environments due to atmospheric $OH\cdot$ recycling mechanisms triggered by isoprene (Taraborrelli et al., 2012; Martinez et al., 2010; Fuchs et al., 2013). Further it was shown that ozonolysis is crucial for HOM formation (Ehn et al., 2014; Kirkby et al., 2016). Another proposed possibility for isoprene suppression of nucleation is the deactivation of sulfuric acid cluster growth due to addition of isoprene oxidation products (Lee et al., 2016). However,



HOMs can nucleate without sulfuric acid (Kirkby et al., 2016) and suppression of nucleation by isoprene is observed in pristine environments such as the Amazon (Martin et al., 2010).

Isoprene oxidation by OH· triggers complex peroxy-radical chemistry with a variety of products such as hydroxy-hydroperoxides (ISOPOOH), hydroperoxy-aldehydes (HPALD) as well as second-generation low-volatility compounds (Teng et al., 2017; Berndt et al., 2016). Isoprene oxidation products with low volatility such as dihydroxyepoxides (IEPOX)
contribute to secondary organic aerosol formation (Carlton et al., 2009; Krechmer et al., 2015; Paulot et al., 2009; Surratt et al., 2010; Lin et al., 2011; Budisulistiorini et al., 2013). However, the interaction of isoprene and monoterpene oxidation chemistry and the consequent effect on nucleation and growth of new particles remains unclear. One consequence of this is an over-prediction of cloud condensation nuclei (CCN) in the Amazon by models that simulate pure biogenic nucleation, but neglect the role of isoprene in new-particle formation (Gordon et al., 2016).

Here, we present experiments performed under atmospherically relevant conditions at the CERN CLOUD chamber and show on a molecular level how isoprene affects the chemistry of monoterpene oxidation, thus reducing nucleation rates as well as early growth rates.

## 2. Methods


The Cosmics Leaving Outdoor Droplets (CLOUD) chamber at the European Center for Nuclear Research (CERN) is a 26.1 $m^3$ stainless steel aerosol chamber, in which a large variety of atmospheric conditions can be recreated under precisely controlled conditions (Kirkby et al., 2011; Kirkby et al., 2016; Duplissy et al., 2016). The chamber is thermally insulated and its temperature can be precisely controlled in the range from -65 °C to 100 °C. In order to reduce contaminations, air mixed
from cryogenic nitrogen and oxygen is used. Trace gases like α-pinene and isoprene can be added and controlled via a two stage dilution system at the parts per trillion by volume level. Mixing is ensured by two magnetically coupled fans. The chamber is equipped with a UV excimer laser and HgXe UV lamps in order to trigger photochemistry. Ion-free conditions can be generated by applying a high voltage electric field across the chamber that sweeps out naturally produced ions (neutral conditions). When this field is switched off, ions produced by galactic cosmic rays penetrating the chamber are
allowed to stay inside the chamber and their effect on nucleation processes can be studied. Using the CERN $\pi^+$-beam increases the ion concentration artificially (see SI Appendix for more detail).

The air inside the chamber is continuously analyzed by a variety of instruments. Organic precursors (α-pinene and isoprene) are measured by a PTR3 instrument (Breitenlechner et al., 2017). HOMs are measured by a nitrate CI-API-TOF (Kürten et al., 2011) that is connected to the chamber via a 1" core sampling probe, where only the inner part of the flow is sampled
into the ion source of the instrument in order to minimize wall losses. Number concentration and size distribution of newly formed particles are measured with an array of butanol based condensation particle counters (CPCs), diethylene glycol based Particle Size Magnifiers (PSMs), as well as a DMA-train and a Scanning Mobility Particle Sizer (SMPS) (see SI Appendix for more detail).

A typical experiment starts with the injection of α-pinene into the particle free chamber (see Fig. S1 and S2), while other
parameters like temperature, humidity and ozone levels are already stabilized. Oxidation of α-pinene by both $O_3$ and OH leads to the formation of HOMs, which subsequently lead to the formation of particles. The experiment is continued without intervention until a steady state in HOMs and nucleation rate has been established. Once the nucleation and growth rates have been determined, the next experiment is performed under slightly different conditions. Parameters that were varied are α-pinene and isoprene levels, ion concentration, UV illumination, sulfuric acid concentration, temperature and relative
humidity.


## 3. Results

We performed several experiments at +5 and +25 °C and relative humidity (RH) ranging from 20 to 80 % with most of the experiments being carried out at 38 % RH. Ozone levels ranged from 30 – 50 ppbv. We directly compare experiments performed with α-pinene as the sole biogenic vapor to experiments with a mixture of α-pinene and isoprene. α-Pinene levels ranged from 0.33 to 2.5 ppbv, while isoprene levels ranged from 2.5 to 10 ppbv. We thus could recreate conditions similar to Kirkby et al. (2016), as well as to regions like the Amazon (Martin et al., 2010; Yáñez-Serrano et al., 2018) and southeastern parts of the United States (Lee et al., 2016).

Ozone attack to the endocyclic α-pinene C=C double bond leads to the well-described formation of highly-oxygenated $RO_2\cdot$ radicals via intramolecular H-shift and autoxidation (mainly $C_{10}H_{15}O_{4,6,8,10}$, from now on referred to as $RO_2(\alpha p)$) as well as a wide spectrum of closed-shell monomers (mainly $C_{10}H_{14,16}O_{5,7,9,11}$) and covalently bound dimers (mainly $C_{20}H_{30}O_{8-16}$ and $C_{19}H_{28}O_{7-11}$, see Fig. 1A) (Ehn et al., 2014; Kirkby et al., 2016; Rissanen et al., 2015; Berndt et al., 2018b; Molteni et al., 2019). These highly-oxygenated organic molecules (HOMs) nucleate at atmospherically relevant concentrations with the help of ions but without other species (e.g. sulfuric acid or bases) required (Kirkby et al., 2016). Here, we group the HOMs according to carbon atom number and define $C_5$, $C_{10}$, $C_{15}$ and $C_{20}$ classes as sum of all HOMs with 2-5, 6-10, 11-15 and 16-20 carbon atoms, respectively. This resembles the basic building block unit of a $C_5$ isoprenoid skeleton.

An isoprene/ozone mixture in the CLOUD chamber produces $C_5H_9O_{5-9}$ $RO_2\cdot$ radicals (referred to as $RO_2(ip)$) which terminate to $C_5H_8O_{5-8}$ and $C_5H_{10}O_{5-9}$ monomers and also some $C_{10}H_{18}O_{8-10}$ dimers under UV-illuminated conditions (see Fig. S5 A, B). The $C_5H_9O_{5-9}$ radicals originate presumably from an OH· addition to isoprene and subsequent autoxidation. Under dark conditions, when the only source of OH· is isoprene ozonolysis at 26 % yield (Malkin et al., 2010), we observe only $C_5$ monomers. None of these molecules are able to nucleate under atmospherically relevant conditions despite having an oxygen to carbon ratio (O:C) ≥ 1, which agrees with earlier observations that products from isoprene ozonolysis do not drive significant new-particle formation (Kamens et al., 1982; Riva et al., 2017).

When isoprene is present together with α-pinene and ozone, the HOM chemistry of α-pinene is altered. We observe the appearance of $C_{15}$ and an increase in $C_5$ class molecules compared to α-pinene only conditions as well as a decrease in $C_{20}$ and $C_{10}$ class molecules (see Fig. 1 and S3). Without isoprene, $RO_2(\alpha p)$ can terminate with another $RO_2(\alpha p)$, thus forming either one $C_{20}$ dimer or two $C_{10}$ monomers. Monomers can also be formed by termination with $HO_2$ or unimolecular termination (Rissanen et al., 2015). The presence of $RO_2(ip)$ offers additional termination channels (Berndt et al., 2018a) (see Fig. 2) and acts as an additional loss term for $RO_2(\alpha p)$. Reactions between $RO_2(ip)$ and $RO_2(\alpha p)$ are expected to result in $C_5$ and $C_{10}$ monomers as well as $C_{15}$ dimers. Most importantly, the reduced $RO_2(\alpha p)$ steady state concentrations lead to a reduction of $C_{20}$ class dimers by roughly 50 % (depending on detailed conditions) compared to their level in the absence of isoprene for all studied α-pinene concentrations (see Fig. S3). To our knowledge the only study that presented ambient measurements of HOMs for an isoprene-rich region is from the SOAS campaign (Southern Oxidant and Aerosol Study, Alabama, USA) (Massoli et al., 2018). When comparing our results with this study, we find good qualitative agreement for the distribution of HOMs with strong contributions in the $C_5$ and $C_{10}$ region and lesser contributions in the $C_{15}$ and $C_{20}$ region. We have to caution however that the $C_{15}$ signal in the reported HOM distribution could also be caused by sesquiterpene products. Additionally, the presence of $NO_x$ affects HOM chemistry in Alabama, which also leads to $C_{20}$ reduction (Lehtipalo et al., 2018).

We measured the particle formation rate directly at a 1.7 nm cut-off diameter with a scanning Particle Size Magnifier (PSM) under neutral (high voltage field cage switched on, see SI Appendix for details) and ion conditions (high voltage field cage switched off, allowing for galactic cosmic ray (gcr) ionization in the chamber), further referred to as $J_n$ and $J_{gcr}$ (see SI Appendix for detail). Fig. 3A shows $J_n$ and $J_{gcr}$ plotted against the total HOM concentration (the sum of the $C_5$, $C_{10}$, $C_{15}$ and $C_{20}$ classes) for the α-pinene only case and α-pinene + isoprene. For +5 °C we find good agreement with Kirkby et al. (2016). However, the presence of isoprene and the consequent change in oxidation chemistry reduces $J_{gcr}$ by a factor of two to four





and $J_n$ even more by around one order of magnitude at 5 °C. The suppression is stronger for lower α-pinene concentrations and thus higher values of R (the ratio of isoprene to monoterpene carbon).

The larger gap between $J_{gcr}$ and $J_n$ with isoprene present compared to α-pinene only conditions is direct evidence that
isoprene oxidation products destabilize the nucleating clusters, thus making cluster stabilization through the presence of charge more efficient. This also confirms that $C_{20}$ class molecules are mainly responsible for pure biogenic nucleation (Frege et al., 2018). $C_{15}$ class molecules, which tend to counteract the losses of the $C_{20}$ class, do not prevent a decrease in $J$. Earlier studies have already suggested that $C_{10}$ class molecules do not possess low enough vapor pressure to qualify as Extremely Low Volatility Organic Compounds (Kurtén et al., 2016; Tröstl et al., 2016) and thus do not drive nucleation, leaving $C_{20}$
class molecules as the most likely nucleator molecules. At +25 °C and UV light illumination, we find that nucleation rates of the pure α-pinene system are reduced by a factor of about 2-3 compared to +5 °C. This is a much smaller reduction in nucleation rate compared to, e.g., the inorganic sulfuric acid water system, for which the same temperature increase reduces nucleation rates by around two orders of magnitude (Kirkby et al., 2011) due to an increase in vapor pressure at warmer temperatures. In our organic system, however, accelerated oxidation chemistry counters the effect of higher vapor pressures.
This includes a higher rate of initial oxidation of α-pinene by ozone, as well as a faster autoxidation process, which leads to HOMs with generally higher oxygen content. When we add isoprene at +25 °C with a constant ratio of isoprene to monoterpene carbon (R = 2), we find a reduction in $J_{gcr}$ of around a factor of about 2. Similar to the data at +5 °C where R ranges from 1.6 to 6.5, we expect a stronger decrease for higher values of R. This can be understood as higher isoprene concentrations enhance $RO_2(ip)$ formation, which in turn reduces $C_{20}$ production and subsequent nucleation. R can reach
levels around 15 in the Amazon (Greenberg et al., 2004) and around 26 in Michigan (Kanawade et al., 2011), where we would thus expect an even stronger isoprene effect on nucleation.

Comparing HOM formation and nucleation for three different α-pinene/isoprene settings, we observe that the addition of 2.7 ppbv of isoprene to an α-pinene/ozone mixture (770 pptv and 49 ppbv, respectively) mitigates $C_{20}$ production and reduces $J_{1.7}$ from 3.2 cm$^{-3}$s$^{-1}$ to 0.81 cm$^{-3}$s$^{-1}$ (see Fig. S6). A rough doubling of both the α-pinene and isoprene levels to 1326 pptv and
4.87 ppbv, respectively, increases overall HOM production; however, $C_{20}$ levels and consequently $J_{1.7}$ remain lower than in the original pure α-pinene setting without isoprene. Thus even increasing monoterpene concentrations can lead to lower $J$ values when isoprene is added as well. Additional evidence for the important role of $C_{20}$ is shown in Fig S9: Regressing each individual HOM peak with $J_{gcr}$ gives high coefficients of determination for $C_{20}$ class molecules.

It has been argued that OH· depletion by isoprene is responsible for the absence of nucleation in isoprene-rich environments
(Kiendler-Scharr et al., 2009; Kiendler-Scharr et al., 2012); however, under atmospheric conditions, isoprene induced OH· recycling can lead to undisturbed high OH· levels, which might not be true in chamber experiments (Taraborrelli et al., 2012; Martinez et al., 2010; Fuchs et al., 2013). In our study we also see an OH· depletion effect due to isoprene addition (see Fig. S1 and SI Appendix for detailed discussion). However, if OH· depletion were the reason for suppression of nucleation, an increase of OH· would lead to an increase in the nucleation rate. When we increase OH· levels by switching on UV lights in
the presence of isoprene, this reduces $RO_2(αp)$ further, as well as the $C_{20}$ and $C_{10}$ class molecules, while enhancing the $C_5$ and $C_{15}$ classes (see Fig. S1, S4 and S5 C, D as well as SI Appendix for details). Accordingly, $J$ is also reduced slightly instead of being increased. In the atmosphere with considerable OH· recycling, this effect, and therefore the suppression of new-particle formation, would be even stronger. We can understand this OH· effect by comparing the reactivity of α-pinene and isoprene towards OH· at our given concentrations. For 300 and 1200 pptv the reactivity of α-pinene towards OH· at +5
°C ([αp]·$k_{αpOH}$) is 25.1 and 6.3 times lower, respectively, than the reactivity of 4 ppbv isoprene towards OH· ([ip]·$k_{ipOH}$). At +25 °C these numbers are similar (25.4 and 6.3, respectively). This implies that any additional OH· provided by e.g. UV illumination will favor the formation of additional $RO_2(ip)$ instead of $RO_2(αp)$, thus favoring the formation of $C_{15}$ over $C_{20}$ and consequently reducing nucleation rates. OH· does not enhance nucleation in this chemical system; it suppresses it.

We performed experiments at +25 °C with three different levels of relative humidity (20, 38 and 80 %) to probe the effect of
water on new-particle formation. Changes in humidity do not significantly affect HOM formation and $J_{gcr}$ (see Fig. S7). $J_n$ increased slightly with humidity, showing an increased stabilization of nucleating clusters by water; however, in gcr conditions, this role is fulfilled more efficiently by ions.



We further studied the effect of sulfuric acid on nucleation of an α-pinene/isoprene mixture (about 1300 pptv and 4.5 ppbv, respectively) in experiments with excess ammonia (0.4 - 2.5 ppbv) in order to reproduce typical conditions in the eastern
parts of the United States (Lee et al., 2016). We find that sulfuric acid does not enhance biogenic nucleation up to a concentration of $5 \cdot 10^6$ cm$^{-3}$ (see Fig. S8). This decoupling of biogenic nucleation from low sulfuric acid levels is similar to the pure α-pinene system reported in Kirkby et al. (2016). At sulfuric acid levels higher than $5 \cdot 10^6$ cm$^{-3}$, nucleation rates depend strongly on sulfuric acid levels, which agrees with a wide variety of atmospheric measurements (Kirkby et al., 2016). In the Amazon, sulfuric acid levels are typically in the range of $1$-$5 \cdot 10^5$ cm$^{-3}$ (Kanawade et al., 2011), well below the
threshold value of $5 \cdot 10^6$ cm$^{-3}$. In Alabama this threshold was exceeded only three times in a 45-day measurement period due to transported sulfur plumes, which led to two events of particles growing to larger sizes (Lee et al., 2016). In Michigan, sulfuric acid concentrations are typically in the range of $1 \cdot 10^6$ cm$^{-3}$ (Kanawade et al., 2011). Sulfuric acid is thus not an important contributor to nucleation in the Amazon as well as different regions of the eastern United States.

We measured the growth rates of freshly nucleated particles from 1.3 nm onwards with a scanning Particle Size Magnifier, a
DMA-train and a nanoSMPS (see SI Appendix for details). The change in HOM chemistry caused by concurrent isoprene oxidation reduces the growth rates of particles in the range of 1.3 – 1.9 nm and 1.8 – 3.2 nm roughly by a factor of two (Fig. 3B and 3C). This confirms that $C_{15}$ class molecules have a higher saturation vapor pressure than $C_{20}$ class molecules and are thus less efficient than $C_{20}$ class molecules at causing growth of the smallest particles. Likewise, most $C_{10}$ class molecules are too volatile to contribute significantly to the early stages of growth (Tröstl et al., 2016). For the size range from 3.2 – 8.0
nm and larger, we observed no suppression effect due to isoprene, indicating that molecules smaller than $C_{20}$ are capable of condensing onto larger particles. We find a linear relationship of growth rate vs $C_{20}$ for 1.3 - 1.9 and 1.8 - 3.2 nm, regardless of isoprene presence. For larger sizes the linear relationship is independent of isoprene presence, when plotted against $C_{15}$ + $C_{20}$; this again indicates that $C_{15}$ contributes to growth at larger sizes (Fig. S10). Besides $C_{15}$ and $C_{20}$, however, even lighter and less oxygenated molecules can contribute to particle growth at larger sizes (Stolzenburg et al., 2018). Growth rates at
+25 °C are typically halved compared to +5 °C due to higher saturation vapor pressure of the HOMs (Stolzenburg et al., 2018), which leads to a higher chance of particles being scavenged while growing, even more so in the presence of isoprene.

Fig. 4 shows the formation rate of particles measured at diameters of 1.7, 2.2, 2.5 and 6 nm for gcr conditions and six concentration values (low/mid/high α-pinene mixing ratios with and without isoprene) at +25 °C. We find that due to the reduced growth rates in the presence of isoprene, a moderate reduction of formation rates at 1.7 nm becomes much more
pronounced, while the particles grow to larger sizes. When we compare α-pinene only data (771 pptv α-pinene, 49 ppbv O$_3$) with a mixture (1320 pptv α-pinene, 39 ppbv O$_3$ and 4.9 ppbv isoprene, orange data points in Fig. 4), $J_{1.7}$ is reduced by 45 %, while the corresponding formation rate at 6 nm is reduced by an order of magnitude. The corresponding precursor concentrations are similar to conditions found in e.g. Alabama (Lee et al., 2016). Isoprene can thus drastically reduce the formation of particles larger than 6 nm even at relatively warm temperatures like +25 °C. This growth-rate driven effect
becomes stronger when α-pinene concentrations are reduced. Our measurements agree with observations of small clusters that are unable to grow efficiently, as has been reported for Alabama (Lee et al., 2016) and the Amazon (Wimmer et al., 2018). Increased levels of preexisting aerosols (i.e. condensation sink) can scavenge freshly nucleated particles (Dada et al., 2017); however, due to the reduced initial growth rates, the likelihood for that process at a given condensation sink is increased when isoprene is present compared to α-pinene only conditions.


## 4. Discussion

Pure biogenic nucleation was first described for α-pinene oxidation in the CLOUD chamber (Kirkby et al., 2016). Global
evaluation of this process with the help of atmospheric modeling found an over-prediction of CCN concentrations in the Amazon, leading to speculation about an as yet unaccounted chemical suppression mechanism for new-particle formation involving isoprene (Gordon et al., 2016). With our findings, we provide the molecular understanding for such a mechanism and identify $C_{20}$ class molecules as the main drivers of biogenic nucleation and early growth. This allows us to refine our understanding of biogenic nucleation for isoprene-rich regions, while at the same time large portions of the atmosphere
where biogenic nucleation is very important, remain unaffected by our findings, especially boreal forests (Gordon et al., 2016).



Suppression of new-particle formation by isoprene was previously attributed to competition for OH· radicals during the initial oxidation of VOCs, which was then thought to be followed by independent oxidation pathways (Kiendler-Scharr et al., 2009). Instead we show that the suppression takes place via $RO_2$· radical interactions that strongly couple the oxidation chains of monoterpenes and isoprene. This is significant beyond the α-pinene/isoprene system, as it indicates the interaction of a variety of atmospheric VOCs with monoterpene-derived HOM formation and new-particle formation. Given that $RO_2(\alpha p)$-$RO_2(VOC)$ reaction rates are competitive (see SI Appendix for details), VOCs whose $RO_2$· radicals lead to products that are smaller than $C_{20}$ when reacting with $RO_2(\alpha p)$ (i.e. reduce the ELVOC (Extremely Low Volatility Organic Compounds) fraction in the HOM distribution) are expected to reduce biogenic nucleation and early growth. On the other hand, VOCs that lead to $C_{20}$ class or larger molecules are expected to accelerate both processes. $RO_2$· termination emerges as the critical step in ELVOC formation and subsequently biogenic new-particle formation. The suppression of biogenic new-particle formation by isoprene and potentially other lighter VOCs, $NO_x$ (Lehtipalo et al., 2018) and elevated $HO_2$ concentrations all proceed along the same lines of $RO_2$· termination and subsequent $C_{20}$ reduction, highlighting the importance of $C_{20}$ class molecules for biogenic new-particle formation.

In summary, we find that isoprene interferes with α-pinene HOM chemistry via $RO_2$· peroxy-radical termination. When isoprene is present, fewer $C_{20}$ class molecules are formed, which directly reduces the nucleation rate. We show that $C_{20}$ class molecules act as "nucleator" species. The reduction of nucleation rate becomes stronger with higher isoprene to monoterpene carbon ratio (R), consistent with earlier observations (Kiendler-Scharr et al., 2009); however, in the monoterpene-isoprene chemical system, increased OH· does not enhance nucleation, but, on the contrary, reduces it due to $C_{20}$ class reduction. Biogenic nucleation in the α-pinene isoprene system is not affected by typical concentrations of sulfuric acid found in the Amazon or in eastern parts of the United States. The change in monoterpene HOM chemistry due to isoprene reduces organic growth rates in the 1.3 – 3.2 nm range by around 50 %, which strongly reduces the probability that the smallest, freshly-nucleated particles will survive scavenging as they grow to larger sizes. While other factors can also inhibit nucleation (e.g. $NO_x$ (Wildt et al., 2014) or a high condensation sink (Dada et al., 2017)), isoprene can make the difference between measurable new-particle formation events and their absence under a variety of atmospheric conditions.

**Data Availability:** Data are available by contacting the corresponding author.

**Author Contributions:** M. H., L. D., M. Sim., D. S., A. C. W., L. F., L. R. A., S. A., F. B., S. B., R. B., R. C., A. D., J. Du., I. E.-H., H. F., C. Fr., C. Fu., H. G., M. G., X. H., J. H., V. H., C. K., T. K., A. K., J. Ka., M. L., K. L., T. M. L., J. L., C. P. L., H. L., H. M., U. M., S. M., V. M., B. M., R. L. M., T. M., R. M., H. E. M., W. N., A. O., T. P., V. P., J. P., L. L. J. Q., M. P. R., Y. S., W. S., S. S., K. S., G. S., M. Sip., Y. J. T., R. V., A. L. V., A. V., M. V.-P., M. W., L. W., D. W., R. W., M. X., P. Y., C. Y., Q. Z., X. Z., J. Kir. and A. T. prepared the CLOUD facility and measurement instruments. M. H., L. D., M. Sim., D. S., A. C. W., L. F., L. R. A., S. A., F. B., A. B., S. B., P. S. B., B. B., R. B., D. C., R. C., A. D., J. Du., I. E.-H., H. F., C. Fu., H. G., O. G., M. G., X. H., J. H., C. R. H., V. H., C. K., T. K., K. L., J. L., C. P. L., H. L., U. M., V. M., B. M., R. L. M., T. M., R. M., H. E. M., T. N., W. N., J. P., M. Pa., L. L. J. Q., M. P. R., C. R., Y. S., W. S., S. S., K. S., G. S., C. T., Y. J. T., R. V., A. V., M. V.-P., L. W., D. W., M. X., P. Y., C. Y., Q. Z., X. Z., J. Kir., A. A. and A. T. collected the data. M. H., L. D., M. Sim., D. S., L. F., H. F., V. H., J. Ka., B. M., T. N., E. P., G. S., R. V., M. X. and C. Y. analyzed the data. M. H., L. D., M. Sim., D. S., A. C. W., L. F., J. Do., H. G., A. K., K. L., R. L. M., M. P. R., M. Sip., A. L. V., P. Y., C. Y., N. M. D., J. Kir., U. B., P. M. W., J. Cu., D. R. W., A. H. and M. K. contributed to the scientific discussion. M. H., L. D., M. Sim., D. S., A. C. W., A. K., N. M. D., J. Kir., U. B. and J. Cu. contributed to writing the manuscript.

**Competing interests:** The authors declare no competing interests.

**Acknowledgments:** We thank CERN for supporting CLOUD with technical and financial resources, and for providing a particle beam from the CERN Proton Synchrotron. We thank P. Carrie, L.-P. De Menezes, J. Dumollard, K. Ivanova, F. Josa, I. Krasin, R. Kristic, A. Laassiri, O. S. Maksumov, F. Malkemper, B. Marichy, H. Martinati, S. V. Mizin, R. Sitals, A. Wasem and M. Wilhelmsson for their contributions to the experiment. This research has received funding from the EC Seventh Framework Programme and European Union's Horizon 2020 programme (Marie Skłodowska Curie ITNs no. 316662 "CLOUD-TRAIN" and no. 764991 "CLOUD-MOTION", MSCA-IF no. 656994 "nano-CAVa", MC-COFUND



grant no. 600377, ERC projects no. 692891 "DAMOCLES", no. 638703 "COALA", no. 616075 "NANODYNAMITE", no. 335478 "QAPPA", no. 742206 "ATM-GP", no. 714621 "GASPARCON"), the German Federal Ministry of Education and Research (projects no. 01LK0902A, 01LK1222A 01LK1601A), the Swiss National Science Foundation (projects no. 20020_152907, 200020_172602, 20FI20_159851, 200020_172602, 20FI20_172622), the Academy of Finland (Center of Excellence no. 307331, projects 299574, 296628, 306853, 304013), the Finnish Funding Agency for Technology and
Innovation, the Väisälä Foundation, the Nessling Foundation, the Austrian Science Fund (FWF; project no. J3951-N36, project no. P27295-N20), the Austrian research funding association (FFG, project no. 846050), the Portuguese Foundation for Science and Technology (project no. CERN/FP/116387/2010), the Swedish Research Council Formas (project number 2015-749), Vetenskapsrådet (grant 2011-5120), the Presidium of the Russian Academy of Sciences and Russian Foundation for Basic Research (grants 08-02-91006-CERN, 12-02-91522-CERN), the U.S. National Science Foundation (grants
AGS1136479, AGS1447056, AGS1439551, CHE1012293, AGS1649147, AGS1602086), the Wallace Research Foundation, the US Department of Energy (grant DE-SC0014469), the NERC GASSP project NE/J024252/1m, the Royal Society (Wolfson Merit Award), United Kingdom Natural Environment Research Council grant NE/K015966/1, Dreyfus Award EP-11-117, the French National Research Agency the Nord-Pas de Calais, European Funds for Regional Economic Development Labex-Cappa grant ANR-11-LABX-0005-01).



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



550 **Figures**

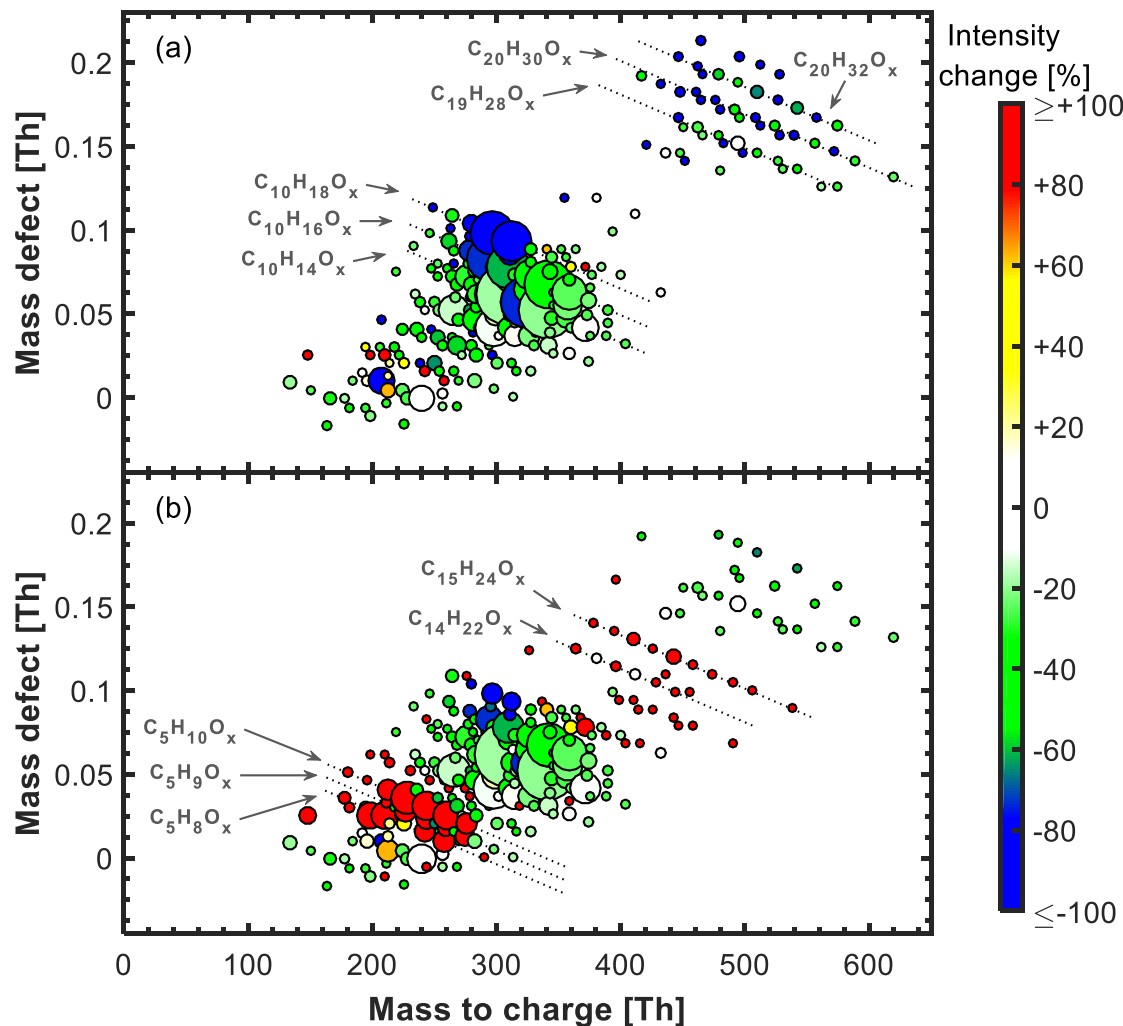

**Figure 1. Mass defect plots of neutral HOM molecules measured with nitrate CI-APi-TOF without isoprene (a) and with isoprene added (b) at +25 °C.** α-Pinene levels were 771 and 1326 pptv, respectively. Ozone levels were 49 and 39 ppbv, respectively. Isoprene was 4.9 ppbv. in (b). Relative humidity was 38 % in (a) and (b). The area of the marker points is linearly scaled with the intensity of the HOM signals. Color code shows the relative intensity change for each HOM peak due to isoprene addition, i.e. the percentage intensity change between (a) and (b). The color for each peak is thus the same in (a) and (b). HOM intensity in (a) was scaled up linearly by 38 % to match [α-pinene]·[$O_3$] levels present in (b) to calculate the intensity change.



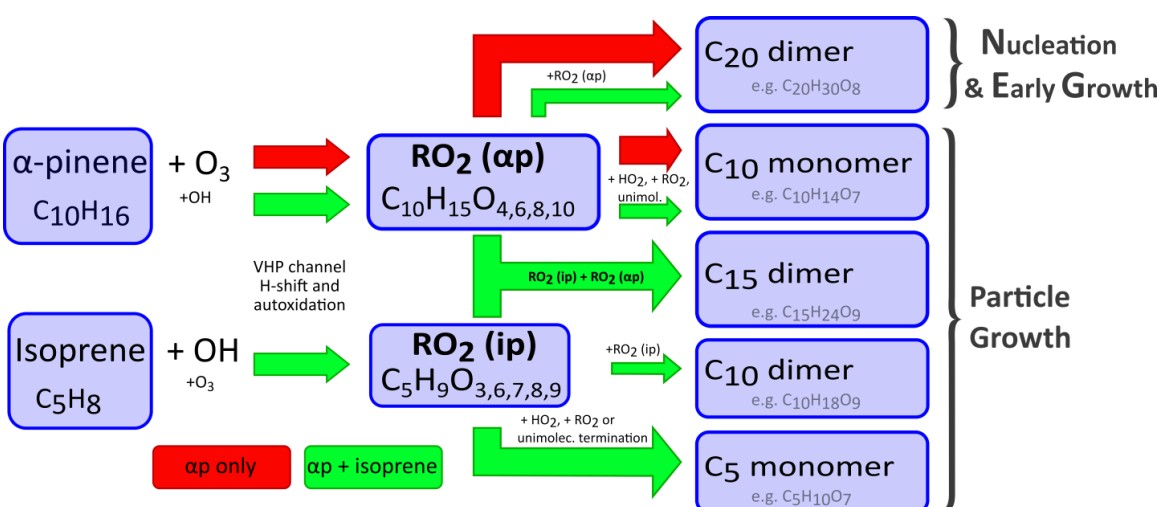

**Figure 2: Proposed mechanism for the interference of isoprene in α-pinene oxidation chemistry.** The pathway of HOM formation of an α-pinene/ozone mixture alone is indicated by red arrows. When isoprene is present, the green arrows indicate the additional interference of isoprene in α-pinene oxidation chemistry via $RO_2\cdot$ radicals. The oxidation of α-pinene at the conditions used in our experiments is dominated by ozonolysis. After the initial ozone attack a $C_{10}H_{15}O_4$ peroxy-radical forms via a vinylhydroperoxyde channel (VHP), which can undergo various intramolecular H-shifts and autoxidation steps. Thus the chain of $RO_2(\alpha p)$ mostly consists of $C_{10}H_{15}O_{4,6,8,10}$. These radicals can terminate either via reaction with other $RO_2\cdot$ radicals, via reaction with $HO_2$ or via unimolecular processes. The resulting closed shell products are then either covalently bound $C_{20}$ class dimers, which are mostly responsible for nucleation or $C_{10}$ class monomers. Possible fragmentation might also lead to a low amount of $C_5$ and $C_{15}$ class molecules being formed even without isoprene present. Isoprene oxidation is dominated by reactions with $OH\cdot$ in the CLOUD chamber, which produce a series of $C_5$ $RO_2\cdot$ radicals ($C_5H_9O_{3,6,7,8,9}$). These $RO_2(ip)$ radicals can now interfere in the termination of $RO_2(\alpha p)$. The reaction of $RO_2(ip)$ with $RO_2(\alpha p)$ can lead to $C_{15}$ class dimers, $C_{10}$ class monomers or $C_5$ class monomers. The reaction of $RO_2(ip)$ with another $RO_2(ip)$ can lead to $C_{10}$ class dimers or $C_5$ class monomers. The presence of $RO_2(ip)$ reduces the steady state concentration of $RO_2(\alpha p)$, as it acts as an additional sink for $RO_2(\alpha p)$. This directly reduces the formation of $C_{20}$ class dimers, as two $RO_2(\alpha p)$ radicals are needed to form one $C_{20}$ class dimer. We link this reduction of $C_{20}$ class dimers to the reduction of biogenic nucleation and early growth rates in the presence of isoprene.



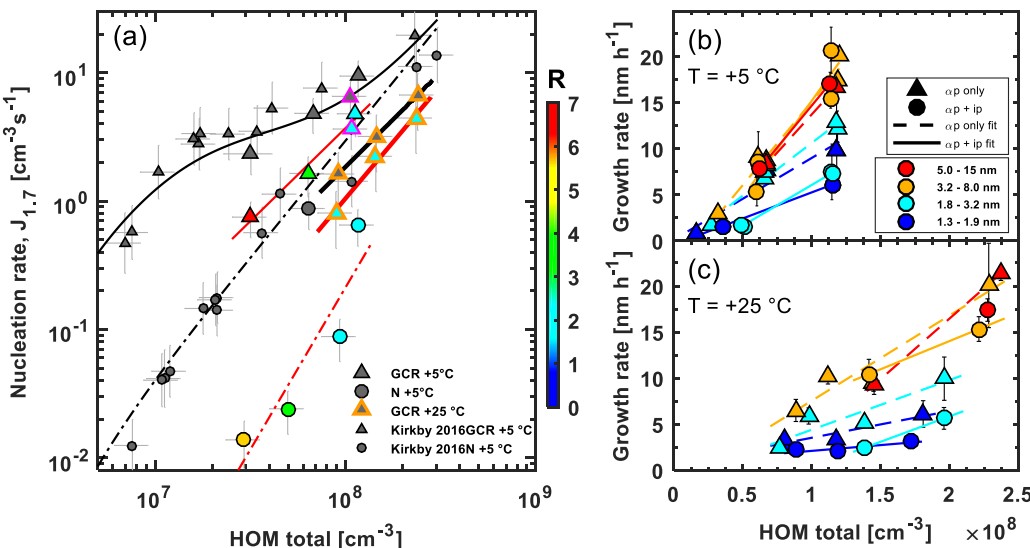

**Figure 3. Pure biogenic nucleation rates at 1.7 nm diameter (a) and growth rates (b, c) against total HOM concentration with and without isoprene added at +5 and +25 °C.** HOM total is defined as the sum of $C_5$, $C_{10}$, $C_{15}$ and $C_{20}$ carbon classes. Relative humidity is 38 % for all data points. **(a)** Triangles represent $J_{gcr}$ and circles $J_n$. Small grey points were taken from Kirkby et al. (2016). Magenta edges indicate UV-illuminated conditions at +5 °C, at +25 °C all data points are with UV light on. Color shows isoprene to monoterpene carbon ratio (R). Black solid and dash-dotted lines are parametrizations of $J_{gcr}$ and $J_n$ from Kirkby et al (2016). Red solid and dash-dotted lines are power law fits to $J_{gcr}$ and $J_n$ in the presence of isoprene at +5 °C. Thick solid black and red line represent power law fits to +25 °C data for α-pinene only and α-pinene + isoprene systems. Bars indicate 1σ run-to-run uncertainty. The overall systematic scale uncertainty of HOMs of +78 %/-68 % and of $J$ for ±47 % is not shown. In **(b)** and **(c)**, triangles represent α-pinene only, circles α-pinene + isoprene conditions. Marker color indicates the size range in which growth rate was measured: dark blue 1.3 – 1.9 nm (measured by scanning PSM), light blue 1.8 – 3.2 nm, orange 3.2 – 8.0 nm (both measured by DMA-train) and red 5.0 – 15 nm (measured by nanoSMPS). Bars indicate 1σ uncertainties in growth rate estimation. Dashed lines are linear fits to α-pinene only data points; solid lines are linear fits to α-pinene + isoprene conditions, respectively.

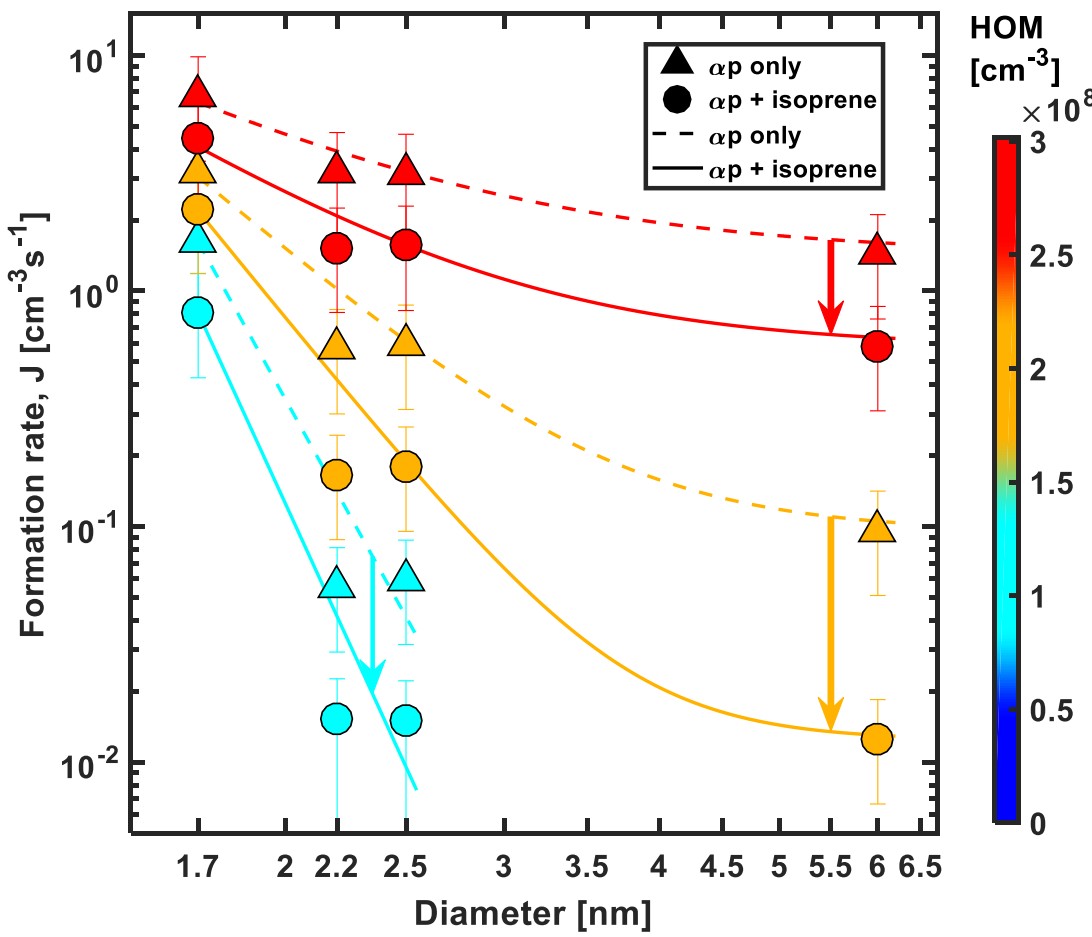

**Figure 4: Formation rate (gcr) vs diameter of particles at +25 °C and 38 % RH.** Triangles represent α-pinene only, circles α-pinene + isoprene conditions. α-Pinene levels were 456, 771 and 1442 pptv for triangles and 677, 1326 and 2636 pptv for circles. Ozone levels were 49 ppbv for triangles and 38 to 40 ppbv for circles. Isoprene levels ranged from 2.7 to 9.8 ppbv for circles. Color code represents HOM concentration. Bars indicate overall scale uncertainty for formation rates of ±47 %. The uncertainty in the diameters is ±0.3 nm. Dashed and solid lines are lines to guide the eye. The steeper slope at lower diameter values is caused by the Kelvin effect, i.e. a smaller growth rate at small sizes that leads to higher losses of newly formed particles. The formation rate measurements at 2.2 and 2.5 nm for the lowest α-pinene/isoprene setting (cyan circles) are upper limits.