# Peer review of "Molecular understanding of the suppression of new-particle formation by isoprene"

_Atmospheric Chemistry and Physics, 2020_

## Referee Comment (RC1) · Anonymous Referee #1 · 18 Apr 2020

New particle formations (NPF) connected to RO2 radical chemistry from the OH oxidation of the monoterpenes are discussed extensively in recent studies especially after the findings of the autooxidation reactions. The isoprene was found to suppress new particle formation events in forested areas. And this is firstly explained by the suppression of the OH due to the addition of isoprene to the reaction system of monoterpene. Nevertheless, the OH concentration was found to be sustained or even enhanced within the isoprene oxidations. So other explanations for the isoprene suppression effects on the NPF are required. As indicated by McFiggans et al., Nature, 2019 isoprene can actually suppress the SOA yield through scavenging of both the OH and the Monoterpenes derived RO2. In this study the authors followed the arguments on that direction and delivered molecular explanations of the isoprene suppression effects on the NPF

based on chamber (CLOUD) experiments and direct measurement of the RO2-HOMs through a nitrate CIMS. I think this paper is in general well written and fit the scope of ACP. It is definitely worth to be published in ACP.

I have the following comments for the authors to consider before publication.

1. It is important for the authors to have a better estimation (e.g. probably through a box model) of the OH and HO2 concentrations in the chamber. As pointed out by the authors, the addition of isoprene will change OH, HO2 and produce isoprene-RO2. It is important to answer whether the suppression of NPF by isoprene is due to the enhanced HO2 or additionally produced RO2. The H-shift of isoprene-RO2 also produce significant amount of HO2.

2. The figure 2 show the scheme of the isoprene impact on the C20 dimer clearly. Nevertheless, it is not clear that what is the branching ratio of the H-shift of isoprene-RO2 proceed to C5H9O7,8,9 and the H-shift yield HAPLDs. In the observations (Figure S1), the observed C5H9O8,9 is presented. Have you also observed C5H9O3,5,7? I understand that the PTR3 can also detect RO2-HOMs. Do the authors also analyzed the RO2-HOMs from the PTR3 signal?

---

## Referee Comment (RC2) · Anonymous Referee #2 · 18 Apr 2020

This paper investigates the molecular mechanisms for the suppression of new particle formation from monoterpenes by isoprene. The authors found that isoprene significantly suppresses the nucleation and the growth of the smallest particles from $\alpha$-pinene oxidation, and showed that this suppression is mainly a result of interference of isoprene oxidation on the production of $\alpha$-pinene HOM dimers, which are the major ELVOCs driving particle nucleation and early growth. This paper is nicely written and provide important molecular constraints on new particle formation in isoprene-rich regions. I recommend the publication of the paper in ACP after the authors address a few minor comments detailed below.

L101-102: The recent study by Berndt et al. (ES&T, 2018) that was cited in this paper reported the interactions between isoprene- and $\alpha$-pinene-derived peroxy radi-

[Figure]

cals in $\alpha$-pinene/isoprene mixed systems. More recently, the Nature paper by McFiggans et al. (2019) that was also cited in the paper clearly showed that the concurrent isoprene oxidation largely scavenges $\alpha$-pinene HOM dimers, in addition to scavenging OH radicals, leading to reduced SOA formation from $\alpha$-pinene/isoprene mixtures. Therefore, the statement "the interaction of isoprene and monoterpene oxidation chemistry......remains unclear" needs to be rephrased.

L270-280: A discussion of the relevant findings in McFiggans et al. (2019) should be included in this paragraph.

L295: "$\alpha$-pinene isoprene system" –> "$\alpha$-pinene/isoprene system".
* * *

---

## Author Comment (AC1) · 8 Jul 2020

The comment was uploaded in the form of a supplement:
https://www.atmos-chem-phys-discuss.net/acp-2020-51/acp-2020-51-AC1-supplement.pdf